# Effects of *Sitobion avenae* Treated with Sublethal Concentrations of Dinotefuran on the Predation Function and Enzyme Activity of *Harmonia axyridis*

**DOI:** 10.3390/insects16070671

**Published:** 2025-06-27

**Authors:** Shaodan Fei, Jiacong Sun, Xingping Ren, Haiying Zhang, Yonggang Liu

**Affiliations:** Institute of Plant Protection, Gansu Academy of Agricultural Sciences, Lanzhou 730070, China; feishaodan@163.com (S.F.); sunjiacong2023@163.com (J.S.); 16634253306@163.com (X.R.)

**Keywords:** *Harmonia axyridis*, *Sitobion avenae*, predation function, acetylcholinesterase, insect detoxification enzymes

## Abstract

This work investigated the effects of aphids treated with sublethal concentrations of dinotefuran on the predation function and enzyme activity of *Harmonia axyridis*. This study assessed the pesticide’s toxicity to aphids and *H. axyridis* using the leaf dipping method and residual film in glass tubes. The results demonstrate that the ingestion of insecticide-treated aphids led to changes in the predatory performance of *H. axyridis*, as well as modulation of the detoxification enzyme activities, which were either upregulated or downregulated. These findings highlight the impact of control strategies using dinotefuran to manage aphid behavior based on natural enemies and establish a foundation for the integrated application of chemical and biological control methods.

## 1. Introduction

The aphid is a significant agricultural pest commonly located on wheat and grass weeds [1]. It mainly induces malnutrition, chlorosis, and wilting in plants by extracting sap and may disseminate the wheat yellow dwarf virus via honeydew [2]. Its adverse effects not only impede the growth and development of crops, resulting in diminished yields, but may also compromise the quality of wheat [3]. In extreme instances, it may result in diminished crop output across the entire field, significantly affecting the agricultural economy [4].

*Harmonia axyridis* is a significant predatory species that reproduces rapidly, exhibits strong adaptability, and possesses a broad range of predatory capabilities against aphids, mites, and other small insects [5]. Consequently, it plays an important role in the biological control of pests, particularly aphids [6]. The utilization of *H. axyridis* for biological control might significantly diminish the aphid population density and decrease reliance on chemical pesticides, thus mitigating environmental contamination and residual pesticide risks in agricultural products [7].

Currently, the management and regulation of aphids predominantly rely on chemical pesticides, with neonicotinoid insecticides being the most utilized [8]. Dinotefuran, a third-generation neonicotinoid insecticide, has low toxicity [9] and extensive applicability and is effective in the control of aphids. Nonetheless, long-term exposure to pesticides may have direct or indirect impacts on the growth and development of natural enemy insects [10]. Research indicates that the hatching rates of eggs and larval survival rates of *H. axyridis* are markedly diminished following exposure to neonicotinoid pesticides [11]. Secondly, the utilization of pesticides either stimulates or suppresses the function of detoxifying enzymes in insects. Furthermore, research conducted by Zhi Cheng et al. indicates that pest individuals surviving sublethal levels of thiamethoxam are consumed by natural enemies, hence indirectly influencing the growth and reproduction of these natural enemies [12]. While ladybugs can substantially decrease aphid populations in the field, their predation on sublethal aphids may negatively impact them [13].

Studies demonstrate that the increased application of chemical pesticides at sublethal levels negatively impacts the predatory efficiency and enzymatic functions (such as acetylcholinesterase and detoxifying enzymes) of *H. axyridis* [14]. Acetylcholinesterase and detoxifying enzymes in insects are important for their survival, playing a crucial role in metabolism and the development of resistance [15]. While current research on dinotefuran primarily examines its toxicity to pests and its immediate impact on natural enemies, there is a lack of studies investigating its impacts on natural enemies following the ingestion of pesticide-exposed prey. The judicious and systematic application of neonicotinoid insecticides, together with the conservation and utilization of beneficial insects, is essential for attaining sustainable agriculture and environmental protection [16].

Using natural enemies for pest management is consistent with ecological principles and contributes to environmental protection [17]. Consequently, the side effects of chemical pesticides on non-target organisms must be taken into account [18]. Following the use of pesticides, the remaining pests may be consumed by natural predators, thereby indirectly influencing the proliferation of these natural enemies [19]. This study analyzed the alterations in the predation function and enzyme activity of *H. axyridis* following its consumption of aphids treated with sublethal concentrations of dinotefuran. It further evaluated the potential ecological risks that this compound may pose to natural enemy populations. This will facilitate a comprehensive understanding of the adverse effects of insecticides, support more effective pest management, help preserve the ecological balance, and promote the adoption of sustainable, eco-friendly control strategies.

## 2. Materials and Methods

### 2.1. Samples

Aphids (*Sitobion avenae*) were initially obtained from fields located in Lanzhou City, Gansu Province, China (36.100324° N, 103.688190° E). They were placed in an artificial climate incubator with specific temperature conditions, (18 ± 1) °C at night and (20 ± 1) °C during the day, along with a relative humidity of 60–70% and a photoperiod of 16 h of light followed by 8 h of darkness. These aphids were not exposed to any pesticides and were provided with wheat leaves as their food source for several generations.

### 2.2. Assessment of Indoor Toxicity of Dinotefuran Towards S. avenae

The toxicity was assessed using the leaf dipping method in accordance with the national agricultural standard NY/T1154.6-2006 [20]. Dinotefuran was prepared as a 110 mg/mL stock solution in acetone, which was subsequently diluted with 0.1% Tween-80 to generate five concentration gradients (0.06, 0.12, 0.24, 0.48, and 0.96 mg/mL). Utilize wheat seedlings with third-instar aphids and immerse them in varying concentrations of dinotefuran for 15 s. Subsequently, remove them and promptly absorb the excess liquid using filter paper. Cultivate them hydroponically in a small beaker, repeating each concentration three times with 40 aphids per trial, while establishing control groups (a control was established by immersing healthy wheat leaves with third instar aphids in an acetone solution for 15 s, followed by culturing). Place them in an artificial climate incubator and calculate the mortality rate after 24 h. (An artificial climate incubator is a high-precision apparatus for maintaining continuous hot and cold temperatures. It is equipped with lighting and humidification capabilities and is ideal for the cultivation of insects and small animals). The mortality rate in the control group exceeded 10% and was deemed ineffective; the LC_20_, LC_30_, and LC_50_ along with their 95% confidence intervals were estimated.

### 2.3. Assessment of Indoor Toxicity of Dinotefuran Towards H. axyridis

Based on the pre-experimental results, dinotefuran was diluted with acetone and formulated into five series of concentration gradients using the half-dilution method: second instar larvae of *H. axyridis* at 0.06, 0.12, 0.24, 0.48, and 0.96 mg/mL; third instar larvae at 0.12, 0.24, 0.48, 0.96, and 1.92 mg/mL; fourth instar larvae and adults at 0.24, 0.48, 0.96, 1.92, and 3.84 mg/mL, alongside control experiments. Add 0.5 mL of the insecticide solution to be tested into each tube, and immediately place the cylindrical tube on a tube rolling machine and rotate it slowly. The insecticide solution is fully contained in a cylindrical glass tube with a flat bottom, measuring 10 cm in length and 2 cm in diameter. As the organic solvent acetone evaporates, it is completely dried to form a thin film tube. The pesticide is evenly distributed on the cylindrical tube, and the cylindrical tube with a drug film is completed. Introduce enough aphids into each tube. Then, place the test ladybug into the pesticide film tube. Cover the opening with gauze and secure it tightly. Finally, position the tube horizontally in a basket. Incubate in a climate-controlled environment, and assess the results after 24 h. The test subjects were composed of second to fourth instar larvae and adult ladybugs. Each treatment is replicated three times, with ten ladybugs for each replication.

### 2.4. Effects of S. avenae Treated with Sublethal Concentrations of Dinotefuran on the Predation Function of H. axyridis

Rear *H. axyridis* until the fourth instar stage, avoiding the use of pesticides. One aphid was placed in each petri dish, covered with film, and pierced with holes. Following a 24 h starving period, they were fed with third instar aphids that had been treated with dinotefuran at LC_20_ and LC_30_ concentrations, with densities of 50, 90, 130, 170, and 210 aphids.

*H. axyridis* with different treatments were reared in an artificial climate incubator, and a control group (fed with untreated aphids) was set up. Each treatment and control group consisted of six replicates. After 24 h, the remaining aphids in the dish were counted, the predation amounts were assessed, and the predatory functional response equation was fitted.

### 2.5. Assessment of Acetylcholinesterase and Detoxification Enzyme Activity in H. axyridis

#### 2.5.1. Collection of Insect Samples

Place one fresh egg of *H. axyridis* into a petri dish, cover it with film, and perforate it with small holes for ventilation. After hatching, the larvae were fed third-instar aphids that had been treated with dinotefuran at concentrations of LC_20_ and LC_30_ for 24 h. The aphids were provided for predation by ladybugs in the following ratios: 45 to first and second instar ladybugs, 120 to third instar ladybugs, and 125 to fourth instar ladybugs (select *H. axyridis* of the same size, with each replication weighing 0.1 g). Samples were collected on the second day following the maturation of the fourth instar ladybugs, promptly frozen in liquid nitrogen, and preserved in an ultra-low temperature freezer (−80 °C) for subsequent analysis.

#### 2.5.2. Assay of Carboxylesterase Activity

The specific method is operated according to the instructions of the Carboxylesterase kit (Beijing Boxbio Science & Technology Co., Ltd., Beijing, China), and then, measure its absorbance value with a spectrophotometer and calculate according to the following formula:CarE (U/g) = 4 × absorbance value (450 nm)/sample mass

#### 2.5.3. Glutathione-S-Transferase Activity Assay

Same as Section 2.5.2. The calculation formula is as follows:GST (U/g) = 0.23 × absorbance value (340 nm)/sample mass

#### 2.5.4. Multifunctional Oxidase Activity Assay

Same as Section 2.5.2. The calculation formula is as follows:MFO (U/g) = 0.167 × x/sample mass

### 2.6. Data Analysis

Excel 2019 was employed to compute the mortality figures. Probit regression analysis was conducted on various developmental stages of aphids and *H. axyridis* utilizing DPS 9.01, resulting in the calculation of LC_20_, LC_30_, LC_50_, the 95% confidence interval, and the regression equation. The chi-square test evaluated the goodness of fit between the regression model and mortality, with a non-significant chi-square value (*p* > 0.05) suggesting a satisfactory model fit. The disparities in toxicity between developmental stages were deemed statistically significant by analyzing the 95% confidence intervals and *p*-values of the LC_50_.

The benefit-to-harm toxicity ratio [21] is used to measure the safety of the pesticides to natural enemies. The lower the benefit–harm toxicity ratio, the greater the toxicity to natural enemies and the lower the safety. The benefit-to-harm toxicity ratio is calculated according to the following formula:benefit-to-harm toxicity ratio = (LC_50_ of the pesticide to *H. axyridis*)/(LC_50_ of the pesticide to aphids)

Logistic regression analysis was performed using RStudio 4.3.0 software to determine the type of functional response, following the method described by Juliano, S.A. [22]. The logistic regression model used is as follows:N_a_/N = Prob{Y = 1} = exp(P_0_ + P_1_N + P_2_N^2^ + P_3_N^3^)/(1 + exp(P_0_ + P_1_N + P_2_N^2^ + P_3_N^3^))

The Holling-II disk equation N_a_ = aNT_t_/(1 + aT_h_N) was used to describe the predation function of *H. axyridis*. Because the experiment time was set to 1 d, T_t_ = 1 was set. Then, N_a_ = aN/(1 + aT_h_N) was obtained, where N_a_ is the number of prey, a is the instantaneous attack rate, N is the prey density, and T_h_ is the handling time

The parameters derived from the Holling-II disk equation were utilized to calculate the search effect (S) of *H. axyridis* on wheat aphids using the subsequent formula: S = a/(1 + aT_h_N).

Data processing was conducted using Excel 2019. IBM SPSS Statistics 19 was utilized for the chi-square test and one-way analysis of variance (normality test, homogeneity of variance test, and Games–Howell test).

## 3. Results

### 3.1. Indoor Toxicity Assessment of Dinotefuran on S. avenae

The toxicity of dinotefuran to wheat aphids was assessed using the leaf dipping method, with results presented in Table 1. The regression equation was y = 3.2604x + 8.4963, with a correlation coefficient of 0.9932, a chi-square value of 0.7742, and *p* > 0.05, indicating that the regression model fit the observed data well. The LC_20_, LC_30_, and LC_50_ values of dinotefuran against third-instar aphids were 0.0467, 0.0585, and 0.0847 mg/mL, respectively, and the 95% confidence intervals were 0.0314–0.0565, 0.0450–0.0682, and 0.0792–0.1040 mg/mL, respectively.

In this experiment, the sublethal doses LC_20_ and LC_30_ of dinotefuran were chosen for further investigations.

### 3.2. Indoor Toxicity Assessment of Dinotefuran on H. axyridis

The toxicity of dinotefuran towards various developmental stages of *H. axyridis* was assessed using the residual film method in glass tubes (Table 2). The findings indicated that LC_50_ values of second, third, and fourth instar larvae and adults were 0.0275, 0.0716, 0.2434, and 0.1406 mg/mL, respectively, and the corresponding benefit–harm toxicity ratios were 0.3247, 0.8453, 2.8737, and 1.6599, respectively. The chi-square values of the probit regression models at each stage were not significant (*p* > 0.05), suggesting that the model fit was good. The sensitivity hierarchy of *H. axyridis* to dinotefuran was as follows: second instar > third instar > adult > fourth instar larvae. The 95% confidence intervals for the second instar larvae did not intersect with those of other instars, and the *p*-value analysis indicated significant differences; however, the confidence intervals for the third and fourth instar larvae and adults overlapped, and the sensitivity differences were not significant.

### 3.3. Results of Predation Function

The predatory functional response of *H. axyridis* to third instar nymphs of aphids treated with sublethal doses of dinotefuran was as follows: following the Chi-square test, χ^2^ < χ^2^(0.05, 20) = 31.410, *p* < 0.05, indicating that the theoretical predation amount aligned with the actual predation amount. Furthermore, the results of the logistic regression analysis (Table 3, Figure 1) showed that the parameter P_1_ < 0, suggesting that the Holling-II model accurately represented the predatory functional effect of *H. axyridis* on aphids exposed to sublethal concentrations of dinotefuran.

The results indicated that as the pesticide dosage increased, the predation rate of *H. axyridis* is diminished (Table 4 and Figure 1). Moreover, feeding wheat aphids treated with LC_20_ and LC_30_ dinotefuran affected the predation function of *H. axyridis*, and the instantaneous attack rate, pest control efficiency, and daily maximum predation amount were reduced by 2.48% and 19.80%, 4.16% and 26.08%, and 1.72% and 7.70%, respectively. On the contrary, the handling time was prolonged by 5.95% and 10.71% (Table 5).

The regulatory effect of natural enemies on prey is associated with their searching efficacy, which is intrinsically linked to the prey density and the natural enemies themselves. The results indicated that, under identical density gradients, the predation efficacy of *H. axyridis* on aphids treated with LC_20_ and LC_30_ dinotefuran exhibited a declining trend relative to the control group. (Table 6 and Figure 2).

### 3.4. Enzyme Activity Results

Following the consumption of LC_20_- and LC_30_-dinotefuran-treated aphids, the carboxylesterase levels in *H. axyridis* were not substantially different from the control, measuring 0.97- and 0.94-fold lower than that of the control, respectively. Glutathione-S-transferase (GST) exhibited an induction effect relative to the control, measuring 1.96- and 1.47-fold higher that of the control, respectively; the induction effect initially increased and subsequently diminished with a rising concentration. The activity of mixed-functional oxidase (MFO) exhibited an induction effect relative to the control, measuring 1.98- and 3.04-fold higher that of the control, respectively. The enzyme activity in LC_30_ was higher than that of LC_20_, suggesting that the induction effect of MFO intensified with higher concentrations (Table 7).

## 4. Discussion

Chemical pesticides utilized in agricultural fields can eliminate the majority of pests. As time progresses and environmental conditions evolve, pesticides degrade into sublethal concentrations, allowing a minority of pests to survive and reproduce, thereby inducing sublethal effects [14]. This study assessed the toxicity of dinotefuran towards *S. avenae* using the leaf dipping method, and the toxicity to *H. axyridis* was studied using the residual film method in glass tubes. The findings indicated that the toxicity of dinotefuran towards various developmental stages of *H. axyridis* was ranked as follows: second instar larvae > third instar larvae > adults > fourth instar larvae. The second-instar larvae exhibited the highest sensitivity, with a significant difference compared to other insect stages; however, no significant difference was observed among the third-instar larvae, fourth-instar larvae, and adults.

Additionally, this research investigated the effects of LC_20_ and LC_30_ dosages of dinotefuran on the daily predation rates of *H. axyridis*. The results demonstrated that the number of aphids consumed by *H. axyridis* for the treated aphids was significantly less than that for the untreated aphids. Wumuerhan et al. [23] found that ladybugs consuming aphids treated with imidacloprid demonstrated diminished prey intake relative to the control group. In contrast, their daily prey consumption increased with the age of the ladybugs [24]. Moreover, the daily predation of *H. axyridis* increased with the augmentation of the aphid density. Upon reaching the saturation of prey intake, the feeding rate stabilized, aligning with the Holling-II functional response. These findings correspond with Kansman, J. T’s conclusion regarding the predation of *H. axyridis* on *Myzus persicae* [25].

The predatory functional response serves as a crucial metric for assessing the predatory efficacy of natural enemies [26]. This research determined that the predatory functional response of *H. axyridis* in both the control and treatment groups conformed to the Holling II-disc equation (Table 5). Y. N. Youn [27] observed that sublethal concentrations of imidacloprid considerably lowered the pest-control ability of *H. axyridis*. The instantaneous attack rate and daily maximum predation amount were lower than those of the control group, and the handling time was greater than that of the control group. This study revealed that the predation function of *H. axyridis* larvae was impacted after consuming wheat aphids treated with LC_20_ and LC_30_ concentrations of dinotefuran. The instantaneous attack rate, daily maximum predation amount, and control efficacy of the fourth instar larvae of *H. axyridis* were reduced compared with the control group. The duration of treatment was extended relative to the control, and the alterations became more pronounced with higher concentrations of the pesticide. The searching effect refers to a predator’s behavioral response to prey. In this study, the searching efficiency of *H. axyridis* on wheat aphids decreased with an increasing prey density, aligning with the findings of Li C et al. [28] on its predation of waterlily aphids. This decrease indicates that higher prey density facilitates prey detection by natural enemies, thereby shortening the search time [29]. Nevertheless, when the aphid density was equivalent, the search efficacy of *H. axyridis* in the dinotefuran-treated group was inferior to that of the control group, indicating the inhibition of the ladybug’s searching capability.

The enhancement of detoxification enzyme metabolism is a significant indicator of the sublethal effects of pesticides on insects. The augmentation of insect metabolic detoxification skills and the reduction in target sensitivity are the principal causes leading to resistance in insect pests. MFOs, GST, and CarE are key detoxification enzymes in pests and play a vital role in the development of insect resistance to various insecticides, including organophosphorus, pyrethroid, and organochlorine compounds [30]. MFOs facilitate the oxidative metabolism of poisonous chemicals in insects while simultaneously boosting the toxicity of these substances through oxidative processes. The experiment demonstrated that *H. axyridis* exhibited enhanced MFO activity after consuming wheat aphids treated with LC_20_ and LC_30_ dinotefuran, in contrast to the control group. This finding aligns with the results of Tianshu Zhang et al., who reported an increase in detoxification enzyme activity in *H. axyridis* following treatment with cyantraniliprole [31]. Nena Pavlidi also discovered that glutathione-S-transferase in *Bemisia tabaci* was induced following treatment with imidacloprid and thiamethoxam [32]. Following the consumption of wheat aphids treated with LC_20_ and LC_30_ dinotefuran, the carboxylesterase levels in *H. axyridis* showed no significant difference compared to the control, exhibiting only a slight inhibitory effect. This limited response may suggest that *H. axyridis* primarily metabolizes dinotefuran through alternative pathways, such as multifunctional oxidases or glutathione S-transferases, thus reducing its reliance on carboxylesterase.

There are significant differences in the direct and indirect effects of dinotefuran on insects. Direct effects usually manifest as acute toxicity or behavioral changes [33]. However, indirect effects are transmitted through the food chain, leading to more complex sublethal consequences on the physiological, behavioral, and ecological functions of insects [34]. This disparity may be attributed to the pesticides’ metabolic pathway, the insects’ detoxifying capacity, and the route of exposure. Consequently, while assessing the influence of pesticides on ecosystems, it is essential to account for both direct and indirect effects. Sublethal doses of dinotefuran influence the predation function and enzyme activity of *H. axyridis*, with the extent of these effects varying across different concentrations. These physiological and behavioral adjustments may represent an adaptive survival strategy in response to dinotefuran-induced stress [33]. Moreover, extensive research has been conducted on the impact of pesticides on the activity of insect detoxification enzymes, yet the findings remain inconsistent. These variations are primarily influenced by factors such as the chemical nature of the pesticide, the target insect species, application concentrations, and treatment protocols [34].

## Figures and Tables

**Figure 1 insects-16-00671-f001:**
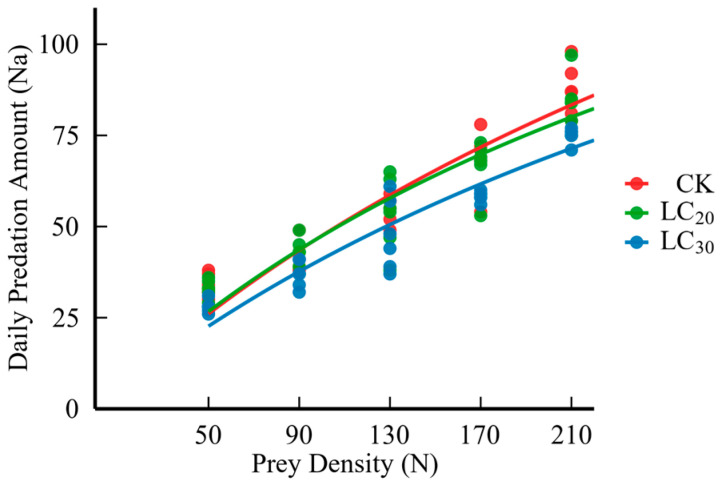
Daily predation of *H. axyridis* on *S. avenae* treated with sublethal doses (LC_20_ and LC_30_) of dinotefuran.

**Figure 2 insects-16-00671-f002:**
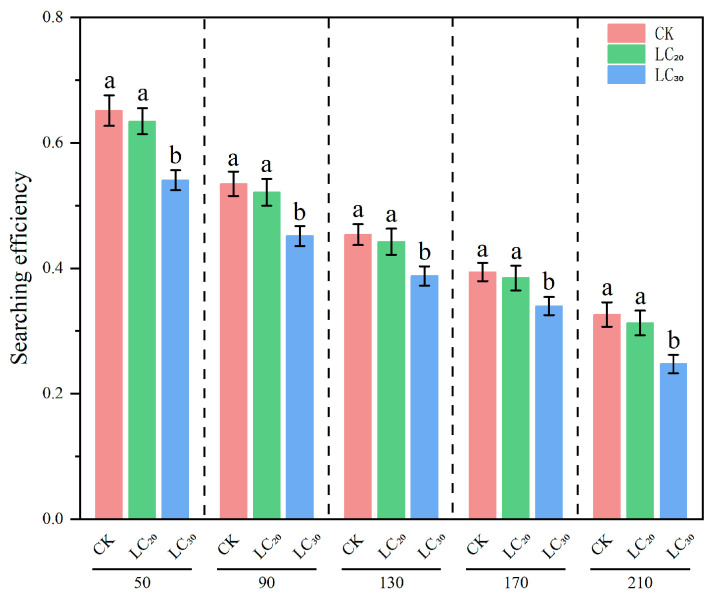
Searching efficiency of *H. axyridis* on *S. avenae* treated with sublethal doses (LC_20_ and LC_30_) of dinotefuran. Notes: different letters at the same density indicate significant differences (Games–Howell test, *p* < 0.05).

**Table 1 insects-16-00671-t001:** Toxicity of dinotefuran against the third instar nymphs of *S. avenae*.

Toxic Regression Equation	Slope	SE	*Chi*-Square Value χ^2^	Correlation Coefficient R^2^	DF	*p*-Value	LC_20_ (mg/mL) (95% Fiducial Limits)	LC_30_ (mg/mL) (95% Fiducial Limits)	LC_50_ (mg/mL) (95% Fiducial Limits)
y = 3.2604x + 8.4963	3.2604	0.707	0.7742	0.9932	3	0.8556	0.0467 (0.0314~0.0565)	0.0585 (0.0450~0.0682)	0.0847 (0.0792~0.1040)

Note: SE: standard error; DF: degree of freedom; LC: lethal concentration.

**Table 2 insects-16-00671-t002:** Toxicity of dinotefuran against *H. axyridis*.

Insect States of *H. axyridis*	Toxic Regression Equation	Slope	SE	*Chi*-Square Value χ^2^	Correlation Coefficient *R*^2^	DF	*p*-Value	LC_50_ (mg/mL) (95% Fiducial Limits)
2nd instar	*Y* = 1.7604*X* + 2.4665	1.7604	0.3411	3.6343	0.9584	3	0.3038	0.0275 (0.0196~0.0404)
3rd instar	*Y* = 1.6272*X* + 1.9817	1.6272	0.3642	2.5113	0.9303	3	0.4733	0.0716 (0.0454~0.1902)
4th instar	*Y* = 2.7274*X* − 1.5085	2.7274	0.4521	1.4219	0.9591	3	0.7004	0.2434 (0.1795~0.4111)
Adult	*Y* = 2.2620*X* + 0.1413	2.2620	0.3294	0.2374	0.9979	3	0.9713	0.1406 (0.1101~0.1921)

Note: SE: standard error; DF: degree of freedom; LC: lethal concentration.

**Table 3 insects-16-00671-t003:** Logistic regression parameters (P_1_, P_2_, P_3_) and functional response type.

Dinotefuran Concentration	P_1_ (N)	P_2_ (N^2^)	P_3_ (N^3^)	Type
0 (CK)	−0.06245	0.000377	−7.411 × 10^−7^	Type II
LC_20_	−0.04876	0.000271	−4.941 × 10^−7^	Type II
LC_30_	−0.03355	0.000170	−2.767 × 10^−7^	Type II

**Table 4 insects-16-00671-t004:** Daily predation of *H. axyridis* on *S. avenae* treated with sublethal doses (LC_20_ and LC_30_) of dinotefuran.

Prey Density (ind./vessel)	Daily Predation Amount (ind.)		
	0 (CK)	LC_20_	LC_30_
50	34.0 ± 1.24 a	33.0 ± 1.00 a	28.0 ± 1.67 b
90	42.8 ± 1.80 a	42.7 ± 1.74 a	37.0 ± 1.48 a
130	54.8 ± 1.51 a	53.7 ± 4.11 a	47.7 ± 3.95 a
170	68.7 ± 3.26 a	66.8 ± 2.90 a	58.3 ± 1.38 a
210	87.3 ± 2.86 a	84.0 ± 3.03 ab	75.0 ± 0.86 b

Note: values in the table are the mean ± standard error; the different letters following the data in the same line indicate significant differences. (Games–Howell test, *p* < 0.05).

**Table 5 insects-16-00671-t005:** Fitting results of the predation functional response model of *H. axyridis* on *S. avenae* treated with sublethal concentrations (LC_20_ and LC_30_) of dinotefuran.

Dinotefuran Concentration	0 (CK)	LC20	LC30
Holling-II equation	N_a_ = 0.8961N_0_/(1 + 0.0075N_0_)	N_a_ = 0.8739N_0_/(1 + 0.0078N_0_)	N_a_ = 0.7187N_0_/(1 + 0.0069N_0_)
Correlation coefficient-r	0.9154	0.9080	0.9239
Instantaneous attack rate-a	0.8961 ± 0.0365	0.8739 ± 0.0123	0.7187 ± 0.0159
Handling time-Th(d)	0.0084 ± 0.0004	0.0089 ± 0.0008	0.0093 ± 0.0007
pest controlefficiency-a/T_h_	107.96 ± 7.76	103.47 ± 9.31	79.80 ± 6.59
Daily maximum predation amount-Na-max(ind.)	119.89 ± 4.87	117.83 ± 9.15	110.66 ± 7.87
Chi-square value-χ^2^	2.715	1.768	1.753

Note: values in the table are the mean ± standard error.

**Table 6 insects-16-00671-t006:** The searching effect equation of *H. axyridis* on *S. avenae* treated with sublethal concentrations (LC_20_ and LC_30_) of dinotefuran.

Dinotefuran Concentration	Searching Efficiency Equation	Correlation Coefficient r
0 (CK)	S = 0.8961/(1 + 0.0075N)	0.9330
LC_20_	S = 0.8739/(1 + 0.0078N)	0.9219
LC_30_	S = 0.7187/(1 + 0.0069N)	0.9304

**Table 7 insects-16-00671-t007:** Effects of sublethal doses of dinotefuran on acetylcholinesterase activity and detoxifying enzymes in *H*. *axyridis*.

Dinotefuran Concentration	CarE (U/g)	GSTs (U/g)	MFOs(U/g)
0 (CK)	112.26 ± 1.7984 a	1.635 ± 0.0093 a	29.227 ± 2.2538 a
LC_20_	108.57 ± 8.2661 a	3.198 ± 0.0136 b	57.867 ± 0.9460 b
LC_30_	105.21 ± 3.3494 a	2.398 ± 0.0094 c	88.837 ± 4.6882 c

Notes: values in the table are the mean ± standard error; the different letters following the data in the same column indicate significant difference. (Games–Howell test, *p* < 0.05).

## Data Availability

The original contributions presented in this study are included in the article. Further inquiries can be directed to the corresponding authors.

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
