# Peer review of "Effects of Sitobion avenae Treated with Sublethal Concentrations of Dinotefuran on the Predation Function and Enzyme Activity of Harmonia axyridis"

_insects, 2025, doi:10.3390/insects16070671_

Round 1
Reviewer 1 Report
Comments and Suggestions for Authors
Review report
General Assessment:
This manuscript presents a well-structured and informative study investigating the effects of sublethal concentrations of dinotefuran on the predatory capacity and detoxification enzyme activity of Harmonia axyridis, an important natural enemy of wheat aphids. The study addresses a significant topic in integrated pest management and provides useful insights into the potential ecological impacts of neonicotinoid insecticides on beneficial predatory insects. However, while the overall design is logical and the experimental approach is relevant, certain aspects of the methodology require revision or further clarification to ensure reproducibility and robustness of the findings.
- The number of replicates is inconsistently reported across different experiments (e.g., predation tests vs. enzyme activity assays). A clear and consistent statement on biological and technical replicates is necessary.
- The sampling procedure for collecting and preparing aphids and ladybugs for biochemical analysis is vague. Details regarding sample pooling, mass per sample, and the number of individuals per replicate should be clarified.
- The leaf-dipping method is described generally, but specifics such as solvent controls, pre-exposure conditions for aphids, and statistical thresholds for defining LC20/LC30 are insufficiently detailed.
- The statistical analysis description is minimal. There is no mention of which software was used beyond Excel 2019, nor how assumptions of normality or homogeneity were verified prior to ANOVA. Confidence interval calculation methods are not specified.
- Terms like “benefit-to-harm toxicity ratio” should be supported with references. Also, equations such as the Holling-II model should include a brief explanation of how parameters were estimated (e.g., nonlinear regression, software used).
- "absorbance value (4×450nm)" is confusing—clarify whether this is a typographical error or a specific calculation method. Ensure all units and concentrations are standardized.
Author Response
Comments 1: The number of replicates is inconsistently reported across different experiments (e.g., predation tests vs. enzyme activity assays). A clear and consistent statement on biological and technical replicates is necessary.
Response 1: Thank you for pointing out the inconsistency in the reporting of replicate numbers across different experiments. For the predation tests, six biological replicates were performed. This higher number of replicates was chosen because each assay involved a single ladybird, and considerable individual variability was expected—such as differences in body size, activity level, feeding capacity, and behavioral responses. Increasing the number of replicates helped to reduce the influence of individual variation and improve the statistical reliability and representativeness of the results. In contrast, the enzyme activity assays were performed using pooled samples from multiple ladybirds per replicate. Since this design inherently reduced biological variability, three biological replicates were deemed sufficient to provide reliable and reproducible results.
Comments 2: The sampling procedure for collecting and preparing aphids and ladybugs for biochemical analysis is vague. Details regarding sample pooling, mass per sample, and the number of individuals per replicate should be clarified.
Response 2: It has been modified as required. Due to slight differences in the size of the ladybugs, the number of ladybugs may vary, but the weight of each replicate was measured at 0.1 grams. (L153-154)
Comments 3: The leaf-dipping method is described generally, but specifics such as solvent controls, pre-exposure conditions for aphids, and statistical thresholds for defining LC20/LC30 are insufficiently detailed.
Response 3: It has been modified as requested. (L121-123, L106-111, L170-173)
Comments 4: The statistical analysis description is minimal. There is no mention of which software was used beyond Excel 2019, nor how assumptions of normality or homogeneity were verified prior to ANOVA. Confidence interval calculation methods are not specified.
Response 4: The modifications have been made according to the requirements. (L170-193)
Comments 5: Terms like “benefit-to-harm toxicity ratio” should be supported with references. Also, equations such as the Holling-II model should include a brief explanation of how parameters were estimated (e.g., nonlinear regression, software used).
Response 5: Thank you for your valuable comment. A relevant reference has been added to support the use of the term “benefit-to-harm toxicity ratio” (see Reference 20 in the revised manuscript).Regarding the Holling-II model, a brief explanation of the parameter estimation method has been included in the Data analysis section (Lines 170–193).
Comments 6:"absorbance value (4×450nm)" is confusing—clarify whether this is a typographical error or a specific calculation method. Ensure all units and concentrations are standardized.
Response 6: This was a typographical error and has been corrected as requested. (L158-168)
Reviewer 2 Report
Comments and Suggestions for Authors
The manuscript by Fei et al. examines how sublethal concentrations of dinotefuran affect the predatory behavior and detoxification enzyme activity of Harmonia axyridis. The results show that after consuming aphids treated with dinotefuran, the ladybird's predation efficiency, attack rate, and handling time were negatively impacted. Detoxification enzyme activity, specifically glutathione-S-transferase and mixed-functional oxidase, increased, indicating a stress response. These findings highlight the potential effects of dinotefuran on natural predators, suggesting caution in its use for pest management. However, I have several major concerns and suggestions:
Major Concerns and Suggestions:
- The manuscript should be edited by a native English speaker, as it contains unclear and awkward sentences throughout.
- The manuscript's data analysis lacks reliability, particularly with respect to lethal concentration calculations and functional response analysis. I suggest the authors consult with a biostatistician or an experienced scientist for guidance on these analyses.
- The resolution of the figures is insufficient, and they are not in an acceptable format for publication.
- Several sections of the manuscript lack a consistent scientific tone and proper use of technical terms. This should be addressed throughout the manuscript.
- There are several miscitations or missing relevant citations. The authors should ensure proper citation of pertinent literature.
- All species names should be italicized throughout the manuscript.
Specific Comments:
L46: Which aphid species are being referred to? Please modify the sentence to: "Aphids are significant agricultural pests commonly found on wheat and grass weeds."
L53-55: This sentence is awkward and should be rewritten in a more formal, scientific tone. For example: "Harmonia axyridis is a significant predatory species that reproduces rapidly, exhibits strong adaptability, and possesses a broad range of predatory capabilities against aphids, mites, and other small insects."
L62: Low toxicity for which species? Do you mean low toxicity for non-target insects?
L62: What is the publication date for Zi Cheng et al.?
L75: Mueller (2018) is a review paper, not directly related to H. axyridis. Please cite studies that are specific to this ladybird species.
L77-80: The authors should cite earlier studies on the immediate side effects of dinotefuran on parasitoids and/or predators.
L86: Please replace "creatures" with "organisms."
L103-107: These sentences should be rewritten to match the current format. There is no need to mention specific chemicals and equipment unless they are directly relevant to the methodology.
L109-115: The expression "mother liquid using acetone" is not scientifically appropriate. Please rewrite the sentence as: "The toxicity was assessed using the leaf dipping method, in accordance with the national agricultural standard NY/T1154.6-2006. Dinotefuran was prepared as a 110 mg/mL stock solution in acetone, which was subsequently diluted with 0.1% Tween-80 to generate five concentration gradients (0.06, 0.12, 0.24, 0.48, and 0.96 mg/mL)."
L118: What is meant by "artificial climate incubator"? Please provide more specific details.
L126: The term "drug" is inappropriate in this context. Please use a scientific term, such as "insecticide solution."
L113-115: The authors should clarify whether the wheat seedlings or the aphids are immersed in the varying concentrations of dinotefuran, as this is unclear from the current wording.
L127: What is meant by "finger-shaped tube"? Please clarify.
L122-125: What species is being referred to here? This sentence needs revision.
L138-145: Please rewrite this section for clarity and conciseness.
L149-155: These sentences should be rewritten with a more formal scientific tone, as they contain incomplete sentences.
L168-175: This section should be rewritten to improve clarity and readability.
L179: The experiment duration is listed as 24 hours, but the observation time is also stated as "1 day." Please clarify whether these terms refer to the same time span. Also, how was "Tt" calculated in hours?
L182: Please specify that "Th" refers to handling time.
L186: There is no need to mention "indoor." The study is clearly conducted under laboratory conditions, so such expressions should be removed from the manuscript.
L190: To confirm whether there is a disparity, the authors should provide goodness of fit values, chi-square, and p-values.
Table 1: The table should include the slope and its standard errors, along with the goodness of fit parameters. "CL" should be replaced with "95% fiducial limits."
Table 2: Similarly, the slope and standard errors, as well as the goodness of fit parameters, should be included. Replace "CL" with "95% fiducial limits."
L199: There is no need to re-mention method details, such as "residual film glass tubes." The use of glass tubes covered with pesticides is already established as the dry film method.
L209-212: Did the authors perform chi-square comparisons in their data analysis? If so, what was the purpose of the chi-square test? The analysis appears awkward and should be clarified.
Figures 1 and 2: The resolution of these figures should be increased. In their current format, they are not clear and do not provide useful information.
L216: Which wheat aphids are being referred to? Sitobion avenae? This sentence is a repetition of information already stated in the Methods section.
Table 4: The authors should provide standard errors of the means for all functional response parameters (a, Th), and compare them across treatments (control and dinotefuran concentrations) using an appropriate multiple comparison test.
L242: What type of calculation is used in "0.97 and 0.94 times that of control"? The same applies to L243 ("1.96 and 1.47 times"). Is "times" or "fold" the correct terminology?
Table 6: What is the meaning of the letter next to the means? What multiple comparison test was used for the comparisons?
L257-259: The authors did not run any analysis on the LC50 of dinotefuran across the developmental stages of H. axyridis. In its current form, the comparison based on numerical values is not scientifically valid. Please refer to Table 2, where the fiducial limits of the calculated LC50 doses overlap.
L262: Replace "devoured" with a more scientific term, such as "consumed" or "fed on."
L268: Who is Jessica T. Kansman? Please rewrite this sentence in a more formal, scientific tone.
Comments on the Quality of English Language
It should be edited by a native English speaker along with an experienced entomologist. I provided details in my review report.
Author Response
Major Concerns and Suggestions:
Comments 1: The manuscript should be edited by a native English speaker, as it contains unclear and awkward sentences throughout.
Response 1: The revisions have been made as requested.
Comments 2: The manuscript's data analysis lacks reliability, particularly with respect to lethal concentration calculations and functional response analysis. I suggest the authors consult with a biostatistician or an experienced scientist for guidance on these analyses.
Response 2: The modifications have been made according to the requirements. (L170-193)
Comments 3: The resolution of the figures is insufficient, and they are not in an acceptable format for publication.
Response 3: We have modified the manuscript accordingly.
Comments 4: Several sections of the manuscript lack a consistent scientific tone and proper use of technical terms. This should be addressed throughout the manuscript.
Response 4: The suggested changes have been implemented.
Comments 5: There are several miscitations or missing relevant citations. The authors should ensure proper citation of pertinent literature.
Response 5: It has already been revised according to the requirements.
Comments 6: All species names should be italicized throughout the manuscript.
Response 6: The revisions have been made as requested.
Specific Comments:
L46: Which aphid species are being referred to? Please modify the sentence to: "Aphids are significant agricultural pests commonly found on wheat and grass weeds."
- It is Sitobion avenae, and it has been altered as per the request. (L46-47)
L53-55: This sentence is awkward and should be rewritten in a more formal, scientific tone. For example: "Harmonia axyridis is a significant predatory species that reproduces rapidly, exhibits strong adaptability, and possesses a broad range of predatory capabilities against aphids, mites, and other small insects."
-The changes have been made according to the specifications. (L53-55)
L62: Low toxicity for which species? Do you mean low toxicity for non-target insects?
-Yes, low toxicity is for some non-target insects, and we have added a new reference (Reference 9) to support this statement.
L62: What is the publication date for Zi Cheng et al.?
-The publication date for Zi Cheng et al. is 2022. ( Sublethal and transgenerational effects of exposures to the thiamethoxam on the seven-spotted lady beetle, Coccinella septempunctata L. (Coleoptera: Coccinellidae). Ecotox. Environ. Safe. 2022, 243, 114002)
L75: Mueller (2018) is a review paper, not directly related to H. axyridis. Please cite studies that are specific to this ladybird species.
-The revisions have been made as requested; please refer to Reference 14 in the revised manuscript.
L77-80: The authors should cite earlier studies on the immediate side effects of dinotefuran on parasitoids and/or predators.
-The revisions have been made as requested; please refer to Reference 16 in the revised manuscript.
L86: Please replace "creatures" with "organisms."
-The revisions have been made as requested. (L85)
L103-107: These sentences should be rewritten to match the current format. There is no need to mention specific chemicals and equipment unless they are directly relevant to the methodology.
-The revisions have been made as requested. (L102)
L109-115: The expression "mother liquid using acetone" is not scientifically appropriate. Please rewrite the sentence as: "The toxicity was assessed using the leaf dipping method, in accordance with the national agricultural standard NY/T1154.6-2006. Dinotefuran was prepared as a 110 mg/mL stock solution in acetone, which was subsequently diluted with 0.1% Tween-80 to generate five concentration gradients (0.06, 0.12, 0.24, 0.48, and 0.96 mg/mL)."
-The revisions have been made as requested. (L104-107)
L118: What is meant by "artificial climate incubator"? Please provide more specific details.
-The revisions have been made as requested. (L114-116)
L126: The term "drug" is inappropriate in this context. Please use a scientific term, such as "insecticide solution."
-We have modified the manuscript accordingly. (L124, L126)
L113-115: The authors should clarify whether the wheat seedlings or the aphids are immersed in the varying concentrations of dinotefuran, as this is unclear from the current wording.
-We have modified the manuscript accordingly. (L108-113)
L127: What is meant by "finger-shaped tube"? Please clarify.
-Changes have been implemented as requested. (L127-128)
L122-125: What species is being referred to here? This sentence needs revision.
-The species is the Harmonia axyridis, and this sentence has been modified as requested. (L120-124)
L138-145: Please rewrite this section for clarity and conciseness.
-It has already been revised as required. (L137-141)
L149-155: These sentences should be rewritten with a more formal scientific tone, as they contain incomplete sentences.
-The sentence has been rewritten as required. (L149-156)
L168-175: This section should be rewritten to improve clarity and readability.
-It has already been revised as required. (L169-175)
L179: The experiment duration is listed as 24 hours, but the observation time is also stated as "1 day." Please clarify whether these terms refer to the same time span. Also, how was "Tt" calculated in hours?
-It has already been revised as required. (L182-185)
L182: Please specify that "Th" refers to handling time.
-The modifications have been made as required. (L185)
L186: There is no need to mention "indoor." The study is clearly conducted under laboratory conditions, so such expressions should be removed from the manuscript.
-The relevant terms have been removed as required.
L190: To confirm whether there is a disparity, the authors should provide goodness of fit values, chi-square, and p-values.
Table 1: The table should include the slope and its standard errors, along with the goodness of fit parameters. "CL" should be replaced with "95% fiducial limits."
Table 2: Similarly, the slope and standard errors, as well as the goodness of fit parameters, should be included. Replace "CL" with "95% fiducial limits."
-We have modified the manuscript accordingly. (L195-L203, Table 1, Table 2)
L199: There is no need to re-mention method details, such as "residual film glass tubes." The use of glass tubes covered with pesticides is already established as the dry film method.
-The relevant statements have been deleted as required.
L209-212: Did the authors perform chi-square comparisons in their data analysis? If so, what was the purpose of the chi-square test? The analysis appears awkward and should be clarified.
-We have modified the manuscript accordingly. (L189-192,L221-225)
Figures 1 and 2: The resolution of these figures should be increased. In their current format, they are not clear and do not provide useful information.
-Figures 1 and 2 have already been changed as required.
L216: Which wheat aphids are being referred to? Sitobion avenae? This sentence is a repetition of information already stated in the Methods section.
-Yes, it’s Sitobion avenae, and we have removed the redundant information as required.
Table 4: The authors should provide standard errors of the means for all functional response parameters (a, Th), and compare them across treatments (control and dinotefuran concentrations) using an appropriate multiple comparison test.
-The functional response parameters were estimated based on the average predation rate, rather than on each individual predation event. As a result, only a single value is available, which does not permit the calculation of a standard error.
L242: What type of calculation is used in "0.97 and 0.94 times that of control"? The same applies to L243 ("1.96 and 1.47 times"). Is "times" or "fold" the correct terminology?
-The data were derived by dividing the treatment group values by those of the control group. Meanwhile, the correct scientific terminology has been used(L254-262), with specific reference to the paper 'Neutralizing immunity in vaccine breakthrough infections from the SARS-CoV-2 Omicron and Delta variants' published in Cell.
Table 6: What is the meaning of the letter next to the means? What multiple comparison test was used for the comparisons?
-a, b, c represent significance of difference, the same letters represent insignificant differences, different letters represent significant differences(L266-268). Duncan's new multiple range test was used for the comparisons
L257-259: The authors did not run any analysis on the LC50 of dinotefuran across the developmental stages of H. axyridis. In its current form, the comparison based on numerical values is not scientifically valid. Please refer to Table 2, where the fiducial limits of the calculated LC50 doses overlap.
-The revisions have been made as requested. (L275-279)
L262: Replace "devoured" with a more scientific term, such as "consumed" or "fed on."
-The modifications have been made as required. (L282)
L268: Who is Jessica T. Kansman? Please rewrite this sentence in a more formal, scientific tone.
-Kansman, J. T. is the abbreviation for Jessica T. Kansman, the author of reference 21. Additionally, the sentence has been rewritten as required.
Round 2
Reviewer 1 Report
Comments and Suggestions for Authors
The authors have addressed all of my questions satisfactorily. I have no further comments and therefore recommend that the manuscript be accepted for publication. Congratulations to the authors!
Author Response
We are grateful for your approval. Thank you.
Reviewer 2 Report
Comments and Suggestions for Authors
I have reviewed the revised version of the manuscript by Fei et al., along with the authors’ responses to previous comments. While I appreciate the authors' efforts to improve the manuscript, several key issues remain unresolved.
The authors’ explanation—that functional response parameters were derived from average predation rates without individual data points, and thus standard errors cannot be calculated—is not methodologically justified. Standard errors and confidence intervals for attack rates and handling times can and should be estimated using raw predation data and established statistical methods.
I strongly recommend reanalysing the functional response data using appropriate statistical tools. First, the functional response type should be determined following Juliano (2001), who proposed a polynomial logistic regression approach: a type II response is indicated by P₁ < 0, while P₁ > 0 and P₂ < 0 suggest a type III response.
Subsequently, attack rates (a) and handling times (Th) should be estimated from the raw data, along with their standard errors or confidence intervals. Treatment effects must be statistically compared using software such as R (e.g., frair, simaR), SPSS, or SAS to ensure that all conclusions are supported by robust statistical analysis.
At present, the manuscript does not include uncertainty estimates or statistical comparisons for the functional response parameters. Therefore, the conclusions are not sufficiently supported and require revision based on appropriate data analysis.
Reference:
Juliano, S. A. (2020). Nonlinear curve fitting: predation and functional response curves. In Design and analysis of ecological experiments (pp. 159–182). Chapman and Hall/CRC.
Line 192: Duncan’s multiple range test is primarily intended for agronomic data. I recommend using a more appropriate post hoc test for ecological experiments.
Lines 126–128: The term “finger-shaped tube” is unclear and unnecessary. Please revise the sentence to: “The insecticide solution is fully contained in a cylindrical glass tube with a flat bottom, measuring 10 cm in length and 2 cm in diameter.” Please replace other instances of “finger-shaped tube” in the text with “cylindrical tube,” which is more accurate.
Line 137: The verb “cultivate” is not appropriate for insects. Please revise to “rear” or “maintain,” which are standard in entomology.
Line 102: The sentence “Dinotefuran is available in the market” lacks context. Please clarify or remove, depending on its relevance.
Lines 238–240 and 265–268: Revise table footnotes as: “Means in the same row followed by different letters are significantly different (Duncan’s multiple range test, p < 0.05).”
Figure resolutions should be improved as previously suggested, as the figures are still unclear.
Comments on the Quality of English LanguageThe manuscript requires thorough language editing. There are numerous grammatical errors and awkward phrasings that hinder clarity and readability. I recommend professional proofreading to ensure the language meets the standards of the journal.
Author Response
Comments 1: The authors’ explanation—that functional response parameters were derived from average predation rates without individual data points, and thus standard errors cannot be calculated—is not methodologically justified. Standard errors and confidence intervals for attack rates and handling times can and should be estimated using raw predation data and established statistical methods.
I strongly recommend reanalysing the functional response data using appropriate statistical tools. First, the functional response type should be determined following Juliano (2001), who proposed a polynomial logistic regression approach: a type II response is indicated by P₁ < 0, while P₁ > 0 and P₂ < 0 suggest a type III response.
Subsequently, attack rates (a) and handling times (Th) should be estimated from the raw data, along with their standard errors or confidence intervals. Treatment effects must be statistically compared using software such as R (e.g., frair, simaR), SPSS, or SAS to ensure that all conclusions are supported by robust statistical analysis.
At present, the manuscript does not include uncertainty estimates or statistical comparisons for the functional response parameters. Therefore, the conclusions are not sufficiently supported and require revision based on appropriate data analysis.
Reference:
Juliano, S. A. (2020). Nonlinear curve fitting: predation and functional response curves. In Design and analysis of ecological experiments (pp. 159–182). Chapman and Hall/CRC.
Response 1: Thank you for your comment. We have reanalyzed the functional response data using the logistic regression method as described by Juliano (2001). As shown in lines 181–194, Figure 1, and Tables 3–6.
Comments 2: Line 192: Duncan’s multiple range test is primarily intended for agronomic data. I recommend using a more appropriate post hoc test for ecological experiments.
Response 2: Thank you for the suggestion. We have replaced Duncan’s test with the Games-Howell test, which is more appropriate for ecological data. The updated results are shown in Tables 4, 5, and 7.
Comments 3: Lines 126–128: The term “finger-shaped tube” is unclear and unnecessary. Please revise the sentence to: “The insecticide solution is fully contained in a cylindrical glass tube with a flat bottom, measuring 10 cm in length and 2 cm in diameter.” Please replace other instances of “finger-shaped tube” in the text with “cylindrical tube,” which is more accurate.
Response 3: The revision has been made as requested; please refer to lines 124–125 for details.
Comments 4: Line 137: The verb “cultivate” is not appropriate for insects. Please revise to “rear” or “maintain,” which are standard in entomology.
Response 4: The revision has been made as requested.
Comments 5: Line 102: The sentence “Dinotefuran is available in the market” lacks context. Please clarify or remove, depending on its relevance.
Response 5: It has been removed as requested.
Comments 6: Lines 238–240 and 265–268: Revise table footnotes as: “Means in the same row followed by different letters are significantly different (Duncan’s multiple range test, p < 0.05).”
Response 6: We have modified the manuscript in accordance with your suggestion.
Comments 7: Figure resolutions should be improved as previously suggested, as the figures are still unclear.
Response 7: The chart has been redrawn for clarity.
Comments 8: Comments on the Quality of English Language
The manuscript requires thorough language editing. There are numerous grammatical errors and awkward phrasings that hinder clarity and readability. I recommend professional proofreading to ensure the language meets the standards of the journal.
Response 8: The manuscript has been thoroughly revised and polished according to the suggestions.
Round 3
Reviewer 2 Report
Comments and Suggestions for Authors
I have reviewed the manuscript titled "Effects of Sitobion avenae treated with sublethal concentrations of dinotefuran on the predation function and enzyme activity of Harmonia axyridis" for the third round of revisions. I also carefully considered the previous comments and suggestions I provided.
I am pleased to note that the authors have addressed all the concerns and suggestions from the earlier rounds. The revisions are well executed, and the manuscript now presents a clearer and more refined data analysis. The data presentation is improved, and all figures and tables are now clearer and easier to interpret.
Due to the recent improvements, I am confident that the manuscript is now in its final, acceptable form. Therefore, I recommend that the manuscript be accepted for publication.